# Iterative Empirical Game Solving via Single Policy Best Response

**Max Olan Smith**
University of Michigan
mxsmith@umich.edu

**Thomas Anthony**
Deepmind
twa@google.com

**Michael P. Wellman**
University of Michigan
wellman@umich.edu

## Abstract

Policy-Space Response Oracles (PSRO) is a general algorithmic framework for learning policies in multiagent systems by interleaving empirical game analysis with deep reinforcement learning (Deep RL). At each iteration, Deep RL is invoked to train a best response to a mixture of opponent policies. The repeated application of Deep RL poses an expensive computational burden as we look to apply this algorithm to more complex domains. We introduce two variations of PSRO designed to reduce the amount of simulation required during Deep RL training. Both algorithms modify how PSRO adds new policies to the empirical game, based on learned responses to a single opponent policy. The first, Mixed-Oracles, transfers knowledge from previous iterations of Deep RL, requiring training only against the opponent's newest policy. The second, Mixed-Opponents, constructs a pure-strategy opponent by mixing existing strategy's action-value estimates, instead of their policies. Learning against a single policy mitigates variance in state outcomes that is induced by an unobserved distribution of opponents. We empirically demonstrate that these algorithms substantially reduce the amount of simulation during training required by PSRO, while producing equivalent or better solutions to the game.

## 1 Introduction

In (single-agent) reinforcement learning (RL), an agent repeatedly interacts with an environment until it achieves mastery, that is, can find no further way to improve performance (measured by reward). In a multiagent system, the outcome of any learned policy may depend pivotally on the behavior of other agents. Direct search over a complex joint policy space for game solutions is daunting, and so more common in approaches combining RL and game theory is to interleave learning and game analysis iteratively.

In *empirical game-theoretic analysis* (EGTA), game reasoning is applied to approximate game models defined over restricted strategy sets. The strategies are selected instances drawn from a large space of possible strategies, for example, defined in terms of a fundamental *policy space*: mappings from observations to actions. For a given set of strategies (policies),[1] we estimate an *empirical game* via simulation over combinations of these strategies. In iterative approaches to EGTA, analysis of the empirical game informs decisions about the sampling of strategy profiles or additions of strategies to the restricted sets. Schvartzman & Wellman (2009b) first combined RL with EGTA, adding new strategies by learning a policy for one agent (using tabular Q-learning), fixing other agent policies at a Nash equilibrium of the current empirical game.

Lanctot et al. (2017) introduced a general framework, *Policy-Space Response Oracles* (PSRO), interleaving empirical game analysis with deep RL. PSRO generalizes the identification of learning targets, employing an abstract method termed *meta-strategy solver* (MSS) that extracts a strategy profile from an empirical game. At each epoch of PSRO, a player uses RL to derive a new policy by training against the opponent-strategies in the MSS-derived profile.

A particular challenge for the RL step in PSRO is that the learner must derive a response under uncertainty about opponent policies. The profile returned by the MSS is generally a mixed-strategy profile, as in strategically complex environments randomization is often a necessary ingredient for equilibrium. The opponent draws from this mixture are unobserved, adding uncertainty to the multiagent environment.

---

[1]The term *policy* as employed in the RL context corresponds exactly to the game-theoretic notion of *strategy* in our setting, and we use the terms interchangeably.

We address this challenge through variants of PSRO in which all RL is applied to environments where opponents play pure strategies. We propose and evaluate two such methods, which work in qualitatively different ways: *Mixed-Oracles* learns separate best-responses (BR) to each pure strategy in a mixture and combines the results from learning to approximate a BR to the mixture. *Mixed-Opponents* constructs a single pure opponent policy that represents an aggregate of the mixed strategy and learns a BR to this policy. Both of our methods employ the machinery of *Q-Mixing* (Smith et al., 2020), which constructs policies based on an aggregation of Q-functions corresponding to component policies of a mixture.

Our methods promise advantages beyond those of learning in a less stochastic environment. Mixed-Oracles transfers learning across epochs, exploiting the Q-functions learned against a particular opponent policy in constructing policies for any other epoch where that opponent policy is encountered. Mixed-Opponents applies directly over the joint opponent space, and so has the potential to scale beyond two-player games.

We evaluate our methods in a series of experimental games, and find that both Mixed-Oracles and Mixed-Opponents are able to find solutions at least as good as an unmodified PSRO, and often better, while providing a substantial reduction in the required amount of training simulation.

## 2 PRELIMINARIES

An RL agent at time $t \in \mathcal{T}$ receives the state of the environment $s^t \in \mathcal{S}$, or a partial state called an *observation* $o^t \in \mathcal{O}$. The agent then chooses an action $a^t$ according to its *policy*, $\pi : \mathcal{O} \rightarrow \Delta(\mathcal{A})$, effecting the environment and producing a reward signal $r^t \in \mathbb{R}$. An experience is a $(s^t, a^t, r^{t+1}, s^{t+1})$-tuple, and a sequence of experiences ending in a terminal state is an episode $\tau$. How the environment changes as a result of the action is dictated by the environment's *transition dynamics* $p : \mathcal{S} \times \mathcal{A} \rightarrow \mathcal{S}$. The agent is said to act optimally when it maximizes *return* $G^t = \sum_l^\infty \gamma^l r^{t+l}$ ($\gamma$ is a discount factor). In *value-based* RL, agents use return as an estimate of the quality of a state $V(o^t) = \mathbb{E}_\pi \left[ \sum_{l=0}^\infty \gamma^l r(o^{t+l}, a^{t+l}) \right]$, and/or taking an action in a state $Q(o^t, a^t) = r(o^t, a^t) + \gamma \mathbb{E}_{o^{t+1} \in \mathcal{O}} \left[ V(o^{t+1}) \right]$.

A *normal form games* (NFG) $\Gamma = (\Pi, U, n)$ describes a one-shot strategic interaction among $n$ players. When there is more than one agent, we denote agent-specific components with subscripts (e.g., $\pi_i$). Negated subscripts represent the joint elements of all other agents (e.g., $\pi_{-i}$). Each of the players has a set of policies $\Pi_i = \left\{ \pi_i^0, ..., \pi_i^k \right\}$ from which it may choose. Player $i$ may select a *pure strategy* $\pi_i \in \Pi_i$, or may randomize play by sampling from a *mixed strategy* $\sigma_i \in \Delta(\Pi_i)$. At the end of the interaction, each player receives a payoff $U : \Pi \rightarrow \mathbb{R}^n$. An *empirical* NFG (ENFG) $\tilde{\Gamma} = (\tilde{\Pi}, \tilde{U}, n)$ is an NFG induced by simulation of strategy profiles. An ENFG approximates an *underlying game* $\Gamma$, estimating the payoffs through simulation. ENFGs are often employed when the strategy set is too large for exhaustive representation, for example when the strategies are instances from a complex space of policies.

The quality of a strategy profile in an ENFG can be evaluated using *regret*: how much players could gain by deviating from assigned policies. Regret is measured with respect to a set of available deviation policies. The regret of solution $\sigma$ to player $i$ when they are able to deviate to policies in $\overline{\Pi}_i \subseteq \Pi_i$ is: $\text{Regret}_i(\sigma, \overline{\Pi}_i) = \max_{\pi_i \in \overline{\Pi}_i} U_i(\pi_i, \sigma_{-i}) - U_i(\sigma_i, \sigma_{-i})$. Example deviation sets $\overline{\Pi}$ employed in this paper are $\Pi^{\text{PSRO}}$, the strategies accumulated through a run of PSRO, and $\Pi^{\text{EVAL}}$, denoting a static set of held-out evaluation policies. The sum of regrets across all players $\text{SumRegret}(\sigma, \overline{\Pi}) = \sum_{i \in n} \text{Regret}_i(\sigma, \overline{\Pi}_i)$, sometimes called the Nash convergence, is a measure of how stable a solution is.

## 3 METHOD

At each epoch $e$ of the PSRO algorithm a new policy is constructed for each player by best-responding to an opponent profile $\sigma_{-i}^{*, e-1}$ from the currently constructed ENFG: $\pi_i^e \in \text{BR}(\sigma_{-i}^{*, e-1})$. These policies are then added to each player's strategy set, $\tilde{\Pi}_i^e \leftarrow \tilde{\Pi}_i^{e-1} \cup \{\pi_i^e\}$, and the new profiles are simulated to expand the ENFG. Algorithm 1 presents the full PSRO algorithm as defined by Lanctot et al. (2017).

One of the key design choices in iterative empirical game-solving is choosing which policies to add to the ENFG. This was first studied by Schvartzman & Wellman (2009a), who termed it the *strategy exploration problem*. We want to add policies that both bring the solution to the ENFG closer to the solution of the full game and that can be calculated efficiently. In PSRO, the strategy exploration problem is decomposed into two steps: solution and BR via RL. In the solution step, PSRO derives a profile $\sigma^{*,e}$ from the current

empirical game. The method for choosing this profile is termed the *meta-strategy-solver* (MSS), formally a function from empirical games to solution profiles $\text{MSS}:\tilde{\Gamma}^e\rightarrow\sigma^{*,e}$. For example, the MSS might compute a Nash equilibrium of $\tilde{\Gamma}^e$.[2] In the BR step, PSRO generates a new policy via RL (the *response oracle*), training against the target opponent profile computed by the MSS. The choice of MSS and response oracle algorithm constitutes a strategy exploration approach and determines the convergence speed of PSRO.

We observe and address two main problems with the standard version of PSRO (Algorithm 1). First, note that the only thing that changes from one epoch to the next is the other-agent strategy profile $\sigma_{-i}$; the transition dynamics remain the same, and the mixed profile $\sigma_{-i}$ itself likely contains other-agent policies encountered in previous epochs. This suggests a compelling opportunity to transfer learning across epochs; however, the BR calculation step works by training anew. Furthermore, because it is responding to some of the same strategies, the new policy learnt may be similar to the strategies added in previous epochs, and so not the most useful addition to the ENFG. Secondly, the opponent profiles are mixtures (i.e., distribution over opponent policies). This makes the environment dynamics more stochastic from the perspective of the RL agent, making learning more difficult. In Section 3.1 we propose an algorithm that transfers knowledge between iterations, and only trains against the single new opponent. In Section 3.2 we present a second algorithm that avoids responding to similar opponent strategies on subsequent iterations, while also addressing the opponent variance issue and providing scalability to multiple other agents.

## 3.1 MIXED-ORACLES

The first problem we address is that during BR calculation there is a missed opportunity for transferring previously learned information. In each epoch, each player learns a BR to a mixed profile of opponent policies. This mixture typically involves the newly added strategies (one per player) for this epoch, but may also include strategies from previous epochs. Training in previous epochs already captured experience against those strategies, so including them in further training entails some redundancy.

The *Mixed-Oracles* algorithm is a variant of PSRO for two-player games, with a modified BR oracle designed to transfer learning across epochs. This method works by learning and maintaining a collection of BRs to each opponent policy $\Lambda_i^e=\left\{\lambda_i^1,...,\lambda_i^e\right\}$, where $\lambda_i^e$ is the BR to $\pi_{-i}^{e-1}$. During each epoch of Mixed-Oracles, a BR is learned for the single new opponent policy, rather than for the mixed opponent-profile generated by the MSS. A BR to the MSS-generated target mixture is then *constructed* from the collection of BR results for constituent policies in the mixture. Constructing the new policy is done through a general Combine-Responses function that maps a set of policies and a distribution over the policies into a single policy. The resulting policy should approximately aggregate the behavior of the policies.

By reusing learned behaviors from previous epochs, Mixed-Oracles allows us to focus training exclusively on new opponent policies. The key design choice is how to combine knowledge from the BRs to individual policies into a BR to any distribution of said policies. We provide a general description of Mixed-Oracles, where the Combine-Responses method is abstract, as Algorithm 2.

Smith et al. (2020) propose *Q-Mixing* as an approach for constructing policies against any mixture of opponent strategies. The method employs Q-values learned against each individual opponent strategy, and thus can support the transfer sought here. Specifically, Q-Mixing averages the Q-values learned against each opponent policy $\pi_{-i}$, weighted by their likelihood in the opponent mixture $\sigma_{-i}$:

$$Q_i(o_i,a_i\,|\,\sigma_{-i})=\sum_{\pi_{-i}}\psi_i(\pi_{-i}\,|\,o_i,\sigma_{-i})Q_i(o_i,a_i\,|\,\pi_{-i}),\tag{1}$$

where $\psi$ determines the relative likelihood of playing an opponent $\psi_i:\mathcal{O}_i\rightarrow\Delta(\Pi_{-i})$. In this study, we use Q-Mixing as our Combine-Responses, where $\psi$ is the prior over the opponent distribution as given by an MSS.

## 3.2 MIXED-OPPONENTS

We next examine PSRO's objective defined by BR to the opponent profile generated by an MSS. Consider the Rock-Paper-Scissors (R-P-S) game in figure 1. Player 1's initial strategy $\pi_1^1$ mostly plays S, which

---

[2]Use of NE as MSS corresponds (for two-player games) to the popular *double oracle* algorithm (McMahan et al., 2003). A variety of other MSSs have been proposed, and assessing their relative merits is a topic of active research (Balduzzi et al., 2019). In experiments reported here, we adopt NE as the baseline MSS; however, our methods readily apply to any MSS.

**Algorithm 1:** Policy-Space Response Oracles (Lanctot et al., 2017)

**Input:** Initial policy sets for all players $\Pi^0$
Simulate utilities $\tilde{U}^{\Pi^0}$ for each joint $\pi^0 \in \Pi^0$
Initialize solution $\sigma_i^{*,0} = \text{Uniform}(\Pi_i^0)$
**while** *epoch $e$ in $\{1,2,...\}$* **do**
    **for** *player $i \in [[n]]$* **do**
        **for** *many episodes* **do**
            $\pi_{-i} \sim \sigma_{-i}^{*,e-1}$
            Train $\pi_i^e$ over $\tau \sim (\pi_i^e, \pi_{-i})$
        $\Pi_i^e = \Pi_i^{e-1} \cup \{\pi_i^e\}$
    Simulate missing entries in $\tilde{U}^{\Pi^e}$ from $\Pi^e$
    Compute a solution $\sigma^{*,e}$ from $\tilde{\Gamma}^e$
**Output:** Current solution $\sigma_i^{*,e}$ for player $i$

**Algorithm 2:** Mixed-Oracles

**Input:** Initial policy sets for all players $\Pi^0$
Simulate utilities $\tilde{U}^{\Pi^0}$ for each joint $\pi \in \Pi^0$
Initialize solutions $\sigma_i^{*,0} = \text{Uniform}(\Pi_i^0)$
Initialize pure-strategy BRs $\Lambda_i^0 = \emptyset$
**while** *epoch $e$ in $\{1,2,...\}$* **do**
    *# Best respond to each new opponent.*
    **for** *player $i \in [[n]]$* **do**
        **for** *many episodes* **do**
            Train $\lambda_i^e$ over $\tau \sim (\lambda_i^e, \pi_{-i}^{e-1})$
        $\Lambda_i^e = \Lambda_i^{e-1} \cup \{\lambda_i^e\}$
    *# Generate new policies.*
    **for** *player $i \in [[n]]$* **do**
        $\pi_i^e \leftarrow \text{Combine-Responses}(\Lambda_i^e, \sigma_{-i}^{*,e-1})$
        $\Pi_i^e = \Pi_i^{e-1} \cup \{\pi_i^e\}$
    Simulate missing entries in $\tilde{U}^{\Pi^e}$ from $\Pi^e$
    Compute a solution $\sigma^{*,e}$ from $\tilde{\Gamma}^e$
**Output:** Current solution $\sigma_i^{*,e}$ for player $i$.

induces a best response of R. Player 1's second strategy $\pi_1^2$ is an approximate best response to R, scoring 80% against it. The best response to $\pi_1^2$ is P. This leads to an empirical game similar to matching pennies, with player 2 playing an even mixture of Rock and Paper; in particular, player 2's meta-strategy never plays S, despite it being good against both player 1 strategies so far.

If, as in PSRO, player 1 then adds their BR to this meta-strategy, which is P, then their strategy set still does not contain the Nash equilibrium of the game. The problem is that there are already strategies in the empirical game that can defeat player 2's strategies, so the new strategy is not useful. This can be detected by inspecting player 2's Q-functions: $Q_2^1$ has a very low valuation of P, and $Q_2^2$ has a very low valuation of R. Responding to S, which is moderately highly valued by both opponent $Q$ functions, is more relevant. To achieve this, we can mix the opponent strategies through their action-value estimates, rather than their action distributions (Figure 1). The BR to this is R, which more usefully extends player 1's strategy set.[3]

This example motivates our second algorithm: Mixed-Opponents. This method also employs a combination method, but instead of combining results from training against previously encountered opponents, it combines the strategies of the opponent mixture themselves to construct a single new opponent policy as a target for training. We refer to the method for generating a new opponent policy from a mixture of opponents the Combine-Opponents, and it has the same functional form as the Combine-Responses. The generalized Mixed-Opponent algorithm is shown in Algorithm 3. We employ Q-Mixing (Equation 1) as our Combine-Opponents. In contrast to Mixed-Oracles, which uses Q-Mixing to transfer Q-values across epochs, here we apply it to average Q-values to define a variant training objective.

## 3.3 ENVIRONMENTS

We evaluate our algorithms on the Gathering (Perolat et al., 2017) and Leduc Poker (Southey et al., 2005) games, both of which are commonly used in the multiagent reinforcement learning field.

Gathering is a gridworld tragedy-of-the-commons game that is partially observable and general-sum. Agents compete to harvest apples that regrow at a rate proportional to the number of nearby apples. The individual's interest in harvesting conflicts with the group's interest in regrowth creating the dilemma. We investigate two versions of this game that differ in the configuration of the map (apple locations, spawn points, etc.): (1) *Gathering-Small* has all apples in a dense grove central in the map, and (2) *Gathering-Open* is a larger map with many spread-out apple groves. The former environment will be used to force two agents to interact, while the latter will allow the study of interactions between more than two players. Further details and images of the environments are included in Section D.1.

---

[3]In Appendix E, we give precise strategies and Q-values that realize this example.

**Algorithm 3:** Mixed-Opponents

**Input:** Initial policies for all players $\Pi^0$

Simulate $\tilde{U}^{\Pi^0}$ for each joint $\pi \in \Pi^0$

Initialize solutions $\sigma_i^{*,0} = \text{Uniform}(\Pi_i^0)$

**while** *epoch e in* $\{1,2,...\}$ **do**

    **for** *player* $i \in [[n]]$ **do**

        $\pi_{-i} \leftarrow$

            Combine-Opponents$(\Pi_{-i}^{e-1}, \sigma_{-i}^{*,e-1})$

        **for** *many episodes* **do**

            Train $\pi_i^e$ over $\tau \sim (\pi_i^e, \pi_{-i})$

        $\Pi_i^e = \Pi_i^{e-1} \cup \{\pi_i^e\}$

    Simulate missing entries in $\tilde{U}^{\Pi^e}$

    Compute a solution $\sigma^{*,e}$ from $\tilde{\Gamma}^e$

**Output:** Solution $\sigma_i^{*,e}$ for player $i$.

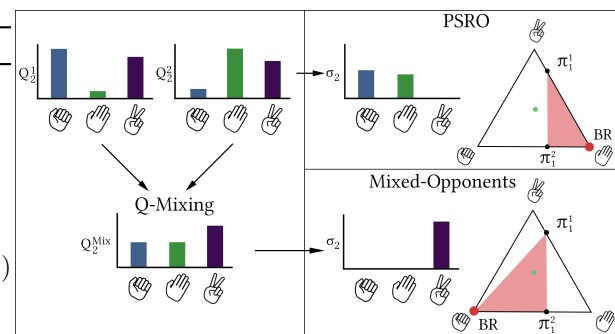

Figure 1: Mixed-strategy opponent targeted in PSRO (top) and corresponding Q-Mixed opponent (bottom). (Left) shows the opponent policy Q-values, and (right) shows the resulting opponent policies and their BRs. In Mixed Opponents, the BR expands the strategy space to include the Nash equilibrium of RPS (green dot), but in PSRO it does not.

Leduc Poker is a reduced version of Poker, designed to capture the important characteristics of a full game of Poker. The game proceeds similarly to Poker, where each player chooses to raise/call/fold through two rounds of betting. In this version, we examine the 2-player versions of Leduc Poker. More detailed information about this environment is included in Section D.2.

## 3.4 HYPERPARAMETERS

The objective of this study is to minimize the number of timesteps spent training a RL agent across all epochs of PSRO. To this end, how timesteps are counted is critical to the comparison of methods. The number of timesteps used for training is typically a hyperparameter for the RL agent. This presents a challenging dilemma, because (1) we must select this before running PSRO, and (2) any particular setting may not work against all opponent-profiles chosen by the MSS.

To address these issues we take a two-step approach (detailed in Section B). First, we select preliminary hyperparameters by searching for their setting that performs best against a random opponent policy. Then we use these hyperparameters to generate a set of opponent-policies (via PSRO) to use as an objective for selecting final hyperparameters. A subset of the generated policies are used as an opponent test-bed to collect two sets of hyperparameters: *pure-hparams* performed best on-average against each individual test-opponent, and *mix-hparams* performed best against a uniform mixture of the test-opponents. These settings are meant to represent the high- and low-variance situations encountered when training mixed and pure strategies.

In this work, our Deep RL algorithm is Double Q-Learning (Hasslet et al., 2016), and our policies have two hidden layers with ReLU activations. 300 hyperparameter settings are sampled in each environment. Complete details are provided in Section D.

## 4 EXPERIMENTS

Regret as a measure depends on the policies that are available for deviation; therefore, it may be misleading to compare across PSRO runs directly. As a more equitable comparison, we instead measure regret when deviating to both the run's discovered policies $\Pi^{\text{PSRO}}$ and a held-out set of evaluation policies $\Pi^{\text{EVAL}}$. The evaluation policies' provide a standard lower-bound on regret across all runs. These policies are constructed by independent PSRO runs, not included in the results here, by sampling from the solution's support. Payoffs are estimated for 30 episodes for 6 policies in all environments, except Gathering-Open with 9.

## 4.1 MIXED-ORACLES

The first research question we address is: does Mixed-Oracles result in a similar quality solution compared to PSRO while utilizing fewer simulation timesteps? We begin by running both algorithms on the Gathering-Small and Leduc-Poker games. We record regrets over PSRO epochs and cumulative training timesteps.

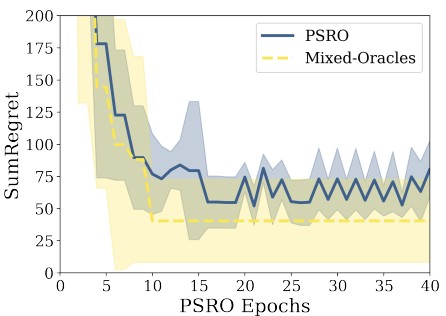 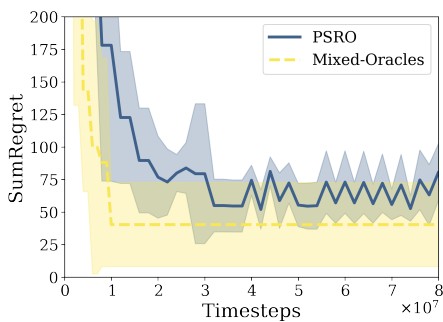

Figure 2: PSRO compared against Mixed-Oracles on the Gathering-Small game. SumRegret over $\Pi^{\text{PSRO}} \cup \Pi^{\text{EVAL}}$ is compared over PSRO epochs (left) and training timesteps (right).

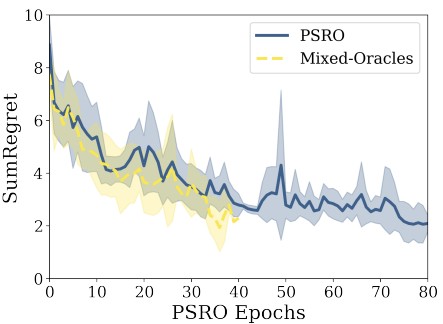 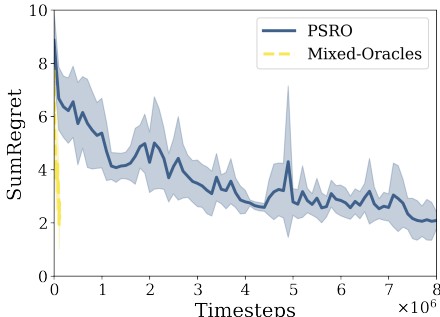

Figure 3: PSRO compared against Mixed-Oracles on the Leduc-Poker game. SumRegret over $\Pi^{\text{PSRO}} \cup \Pi^{\text{EVAL}}$ is compared over PSRO epochs (left) and training timesteps (right).

Figure 2 compares the convergence speed of both algorithms on the Gathering-Small environment. We found that Mixed-Oracles converged in roughly half the number of epochs while utilizing a quarter of the number of simulations. Moreover, the solution discovered by Mixed-Oracles had approximately 30% less regret. Figure 3 shows the result on Leduc-Poker, where we observe Mixed-Oracles finding a comparable solution in dramatically fewer cumulative timesteps. These results confirm our hypothesis that Mixed-Oracles will find comparable solutions to PSRO in fewer training timesteps.

## 4.2 MIXED-OPPONENTS

Next, we address the question: does Mixed-Opponents result in a better quality solution compared to PSRO while utilizing fewer simulation timesteps? The methodology of this experiment follows the previous experiment: measuring SumRegret on both the Gathering-Small and Leduc-Poker games.

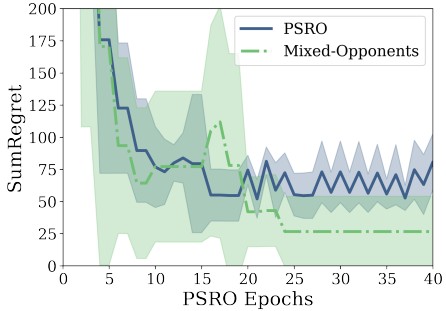 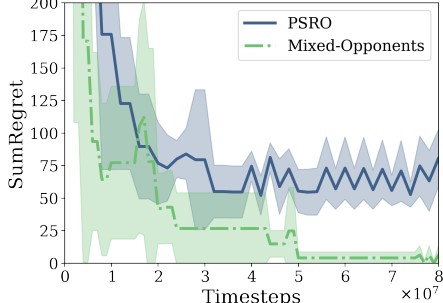

Figure 4: PSRO compared against Mixed-Opponents on the Gathering-Small game. SumRegret over $\Pi^{\text{PSRO}} \cup \Pi^{\text{EVAL}}$ is compared over PSRO epochs (left) and training timesteps (right).

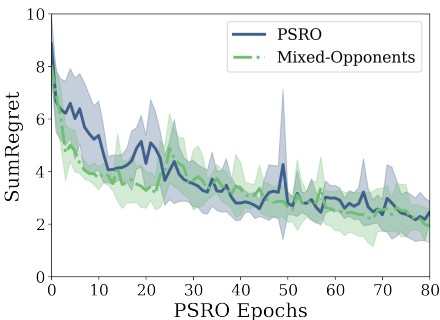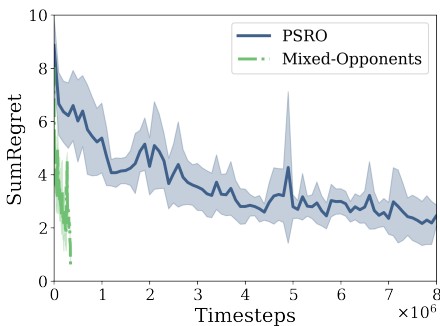

Figure 5: PSRO compared against Mixed-Opponents on the Leduc-Poker game. SumRegret over $\Pi^{\text{PSRO}} \cup \Pi^{\text{EVAL}}$ is compared over PSRO epochs (left) and training timesteps (right).

Figure 4 compares the convergence speed of both algorithms on the Gathering-Open game. Both algorithms appear to converge initially after 25 epochs, and at this point, Mixed-Opponents had discovered a solution with 50% less regret. However, while PSRO remained converged as time went on, Mixed-Opponents continues to improve and nearly solves the game. Figure 5 shows the results on Leduc-Poker. Similar to Mixed-Oracles, we also find that Mixed-Opponents finds a similar solution to PSRO in dramatically fewer timesteps. Together these results confirm our second hypothesis that Mixed-Opponents may find a comparable solution to PSRO in less training simulations.

### 4.3 > 2 PLAYER GAMES

An advantage of Mixed-Opponents over Mixed-Oracles is that it can be applied to games with more than two players. Does Mixed-Opponents reduce *training-simulation* when compared to PSRO in games where there are more than two players? We run both algorithms on the Gathering-Open game with three players. We limit each profile in the ENFG to three simulations, to handle the combinatorial explosion of profiles. Figure 6a shows our results where Mixed-Opponents finds a similar quality solution to PSRO in half of the time.

### 4.4 SHARED HYPERPARAMETERS

An important choice in these experiments has been how to measure timesteps elapsed. We used two sets of hyperparameters: specialized for low- and high-variance outcomes of state, which are a result of facing pure- and mixed-strategy opponents respectively. This was motivated by the assumption that lower variance would require less training. In this section, we question that assumption and ask: do Mixed-Oracles and Mixed-Opponents perform at least as well as PSRO when given the same RL hyperparameters?

To answer this question we run all three algorithms with the same set of hyperparameters, forcing all to adopt the same simulation budget. We report results for the Gathering-Small game in Figure 6b. The trends observed in the previous experiment reoccur: Mixed-Oracles and Mixed-Opponents find solutions at least as good as PSRO in fewer timesteps, confirming our hypothesis. In particular, under the pure-hparams, where simulations are more limited, Mixed-Opponents find a better solution than PSRO in half the simulation timesteps.

## 5 RELATED WORK

The first published paper to explicitly estimate an ENFG from an agent-based simulation was a study by Walsh et al. (2002) on pricing games and auctions. They analyzed symmetric games with up to twenty players over a small number of hand-crafted heuristic strategies. A few additional works in succeeding years explored the approach, which was developed into an algorithmic framework under the EGTA label by Wellman (2006). The strategy exploration problem within EGTA was first identified and studied by Schvartzman & Wellman (2009a). In this work, they explored three heuristic solutions and found that BR was the most effective method. They also illustrated situations where equilibrium response had poor performance.

Before this formalization, Phelps et al. (2006) were the first to utilize automated strategy generation. Their approach is still seen in PSRO today, where iterative BRs to equilibrium are used to build an empirical

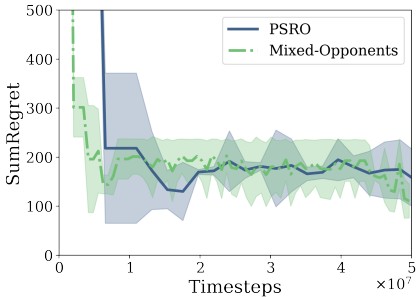 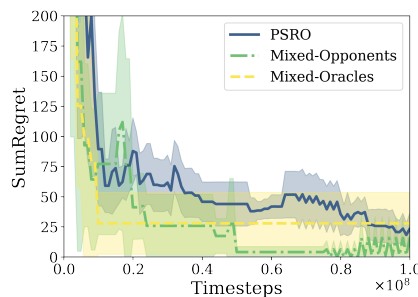

(a) Mixed-Opponents evaluated on the Gathering-Open game with 3 players.

(b) Comparison of algorithms when set to the same Deep RL hyperparameters.

Figure 6: Experiments on extension beyond two players and using shared hyperparameters.

game. They utilized a small parametric policy space and used genetic search as their approximate oracle method. Schvartzman & Wellman (2009b) followed this up by introducing reinforcement learning as the approximate oracle method. These algorithms, alongside a trend of deep reinforcement learning being used to solve difficult games, set the stage for the PSRO framework. PSRO highlighted a subclass of EGTA algorithms where an iterative construction of an empirical game is built through iterative BR to an opponent-profile. At each epoch, the opponent profile is determined by a chosen meta-strategy solver (MSS). The PSRO algorithm builds heavily upon the Double-Oracle (DO) algorithm, which iteratively solves for the Nash Equilibrium in two-player zero-sum sub-games (McMahan et al., 2003).

This has culminated in an interest in discovering good strategy-exploration methods within PSRO. PSRO's strategy exploration happens in two phases: first, a MSS produces an opponent-profile as a learning objective, then reinforcement learning is used as an approximate oracle to best-respond to the objective. Many popular algorithms for computing strategies in game theory have been adapted as MSS: linear-programming, replicator-dynamics Taylor & Jonker (1978), regret-minimization (Blum & Mansour, 2007), and regret-matching (Hart & Mas-Colell, 2000). Moreover, popular game-solving algorithms can also be viewed through the lens of a MSS. Fictitious play (Brown, 1951) is a MSS where the resulting profile is a uniform-strategy. Similarly, self-play's resulting profile is a pure strategy containing only the newest policy.

Prompting recent work to leverage the additional information gleaned through the empirical game to construct new MSSs. Wright et al. (2019) extends DO's MSS to return a decay-weighted linear combination of the current and previous solutions. Omidshafiei et al. (2019) proposes Markov-Conley Chains (MCC) as a solution concept that relates to their new proposed quality measure of policy called $\alpha$-Rank, and Muller et al. (2020) investigated its use as a MSS.

McAleer et al. (2020) also tackles the problem of the compute requirements of PSRO by addressing its runtime. Their Pipeline-PSRO algorithm enables the RL process across epochs to be trained in parallel. The parallel technology could be leveraged to reduce runtime of both Mixed-Oracles and Mixed-Opponents.

## 6 CONCLUSIONS

In this study, we investigate PSRO's approach to strategy exploration and extensions that focus on reducing the time spent on training new policies. PSRO's strategy exploration occurs in two steps: solution and BR via RL. We first propose Mixed-Oracles which approximates the solution step by training separate BRs to each opponent policy and then combining the results into an approximate BR to the solution. This affords our algorithm the ability to transfer learning about previously encountered opponents. Next, we introduce Mixed-Opponents which modifies the BR training by constructing a single opponent policy that represents an aggregate of the mixed strategy and learns a response to this new opponent. Mixed-Opponents also trivially scales to games with more than two players. Both of these methods further mitigate the variance in state outcomes, by removing an unobserved distribution of opponents, easing learning.

ACKNOWLEDGMENTS

Work at the University of Michigan was supported in part by MURI grant W911NF-13-1-0421 from the US Army Research Office.

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
