# OpenReview forum: "Iterative Empirical Game Solving via Single Policy Best Response"
_ICLR.cc/2021/Conference — ICLR 2021 Spotlight_

### Official Review · AnonReviewer2 · 2020-10-27
**Good ideas, but do not seem significant enough**

**Rating:** 7
**Confidence:** 2

**Review:**

The paper suggests two techniques to improve the calculation of empirically figuring out a Nash equilibrium using an iterative application of best-response dynamics. One method learns the best-response to the previously used strategy. The other uses that technique to model the opponent, and then best-responds to the modeled opponent. The experiments show a faster reaching to NE than without these changes.

The paper is well-written and explained, and is accessible even to researchers not well-versed in ML topics. While, the suggested changes are rather straightforward, they do indeed lead to the expected advance (shorter times to reach NE). I was convinced by further introspection that this is a significant enough contribution, to merit acceptance.

Of course, a more significant change to the algorithm, leading not only to a shorter time but to convergence to better equilibria (in cases where multiple exist) would be far more compelling.

---

> ### Author Response · Authors · 2020-11-13
> **Response to Reviewer #2**
>
> Thank you for taking the time to read our work. We are happy to hear you found the work very clear, even for a general audience.
>
> You say that advances that shorten convergence time for PSRO are not significant advances. We disagree: (i) the computational cost of reinforcement learning algorithms is problematic in general. (ii) The PSRO framework uses an independent application of reinforcement learning in its inner loop, each iteration. This multiplies problem (1) by the iteration count.
>
> The result of these problems is that PSRO can be prohibitively expensive to use. An excellent example of both the value of the PSRO framework, but also the particularly high cost of using it, is recent work on StarCraft 2. *Far from being insignificant, addressing the cost of using PSRO is a crucial challenge in imperfect information games research.*
>
> The motivations of our work (identified by reviewer 1: “1. starting anew everytime, 2. slower exploration of strategy space, 3. stochastic dynamics making learning difficult.”) address these issues. By transferring knowledge between iterations of PSRO (motivation 1), we have reduced the multiplicative factor caused by multiple applications of reinforcement learning (problem ii). By exploring the strategy space more efficiently (motivation 2) we reduced the number of applications of reinforcement learning required (alleviating problem ii), and by reducing the stochasticity of learning dynamics (motivation 3) we reduced the cost of applying reinforcement learning (problem i).

---

### Official Review · AnonReviewer1 · 2020-10-28
**Important step towards efficient PSRO**

**Rating:** 7
**Confidence:** 4

**Review:**

The paper focuses on resolving the computational and sample efficiency challenges with current PSRO style approaches. To this end it proposes two different modifications to the standard PSRO setup: 1) Mixed Oracles, and 2) Mixed Opponents. These approaches allow avoiding resetting learning after each outer loop epoch and reduce the stochasticity of dynamics during training. Thee efficacy is demonstrated on relatively simple games but using Deep RL policies where the proposed approaches are at least on par with standard PSRO approach in terms of final performance while drastically improving the sample efficiency.


Although the paper has quite a few typos and unclear writing in places, it has a fantastic setup with clear description of the problems with standard PSRO it's trying to resolve. The paper points out the clear computational deficiencies with current PSRO style approaches: 1) starting anew everytime, 2) slower exploration of strategy space, 3) stochastic dynamics making learning difficult.


The paper is mostly focused on two-player zero sum games.
Approach heavily dependent on efficacy of Q-mixing approach (although the idea is more general), so currently limited to discrete action problems.
Moreover, since there are no alternatives to Q-mixing, we don't actually get any understanding/intuition for why this works and how important is the performance of "TransferOracle" or "OpponentOracle".

Although section 4.4 suggests that the two stage hyperparameter selection wasn't as important, results in Fig 6b are too noisy to fully accept. It's unclear how much compute/samples were required for the hyper parameter selection. In general the results are a lot more noisier for the proposed approaches vs standard PSRO.

**Minor**
- Environment dynamics should be SxA rather than OxA
- Algorithm 4.1 does not exist?
- Fig 1. Is it because of PSRO with Nash. Would a different MSS work differently?
- "proposes MCC as a solution concept": not everyone would know what MCC refers to.

---

> ### Author Response · Authors · 2020-11-13
> **Response to Reviewer #1**
>
> Thank you for taking the time to review our work and offer many suggestions for improvement. We are glad to hear you found our setup and problem description are clear.
>
> First, we would like to address the typos and lack of clarity. We have gone through all of the relevant minor comments and corrected these issues. Moreover, we have since performed additional copy-edit passes of the entire draft cleaning up more minor grammatical errors. We appreciate the explicit list of issues that were provided.
>
> Next, we would like to address the proposed methods’ dependency on Q-Mixing. This is a major limitation of our study, due to the lack of methods functionality similar to Q-Mixing, and as a result, has led to our focus on discrete action games. This leads to two limitations you’ve mentioned: (1) Q-Mixing potentially confounding the benefit gained from Mixed-Oracles or Mixed-Opponents and (2) lack of evaluation on continuous-action games. We believe both of these points are important directions for future work on sample efficient PSRO. This study is meant to serve as a first-step towards addressing all of these diverse problems. We hypothesize that the naive version of Q-Mixing that we employed in this study serves only as an empirical lower-bound on performance gains afforded by Mixed-Oracles and Mixed-Opponents.
>
> Finally, we will soften the claims about the importance of the hyperparameter selection method. We selected 300 hyperparameter configurations from the options listed in the supplemental material. Each configuration was evaluated for both pure-hparam and mix-hparam. We have added the number of configurations to the paper, thank you for pointing this missing detail.
>
> Minor comments:
>  - Fixed the environment dynamics definition.
>  - Fixed to correctly refer to the algorithm instead of a section.
>  - Figure 1 is an artifact of MSS producing a distribution of deterministic policies (rationally taking the best action) and therefore is independent of the solution concept of the MSS.
>  - Fixed to instead directly refer to Markov-Conley Chains.

---

> > ### Comment · AnonReviewer1 · 2020-11-17
> > **Thanks**
> >
> > Thanks for the response. The paper is definitely clearer to read! I would still love to know your thoughts on why mixed-oracles/mixed-opponents are so much more  noisier. Is that an artifact of environment stochasticity closer to better performance or because of Q-Mixing?

---

> > > ### Author Response · Authors · 2020-11-17
> > > **Response to Noise in Results**
> > >
> > > Thank you for the interesting question.
> > >
> > > We suspect that the increase in noise is an artifact of Q-Mixing. The Gathering environments are non-stochastic environments (with perfect information) and still exhibit this phenomenon. The version of Q-Mixing we utilize only constructs an approximate aggregation of the policies because constructing a true aggregation is computationally infeasible (requiring iterating over all state-action pairs). It is not trivially clear how this impacts the noise experienced by the respective PSRO algorithm, and something we are actively investigating.

---

### Official Review · AnonReviewer4 · 2020-10-28
**Review of Iterative Empirical Game Solving via Single Policy Best Response**

**Rating:** 7
**Confidence:** 2

**Review:**

Summary
The paper proposes two new methods in the Policy-Space Response Oracle framework. These approaches permit to reuse past knowledge in order to reduce the amount of data required for the RL training. The first algorithm Mixed-Oracles transfers the previous iteration of Deep RL, instead of the second one, Mixed-Opponents, transfers existing strategy action-value estimates.

Strengths
The paper proposes two convincing alternatives to reuse previous knowledge in the PSRO framework. The two ideas are based on Q-mixing approach: the first one uses this method to transfer Q-values across epochs, the second one to design a new training objective.
The experiments show that the proposed methods find a good solution using less simulation than the original PSRO framework.

Weakness
The paper is not very novel, since it uses previous approaches (PSRO and Q-mixing) to transfer knowledge for the PSRO framework.
I am not aware of recent works in this framework but could be useful to compare the proposed approaches with P2RO [1].
The Mixed-Opponent section needs a better explanation of the use of Q-mixing as a training objective.

[1]  McAleer, S., Lanier, J., Fox, R., & Baldi, P. (2020). Pipeline PSRO: A Scalable Approach for Finding Approximate Nash Equilibria in Large Games. arXiv preprint arXiv:2006.08555.

---

> ### Author Response · Authors · 2020-11-13
> **Response to Reviewer #4**
>
> Thank you for taking the time to review our work and provide suggestions for improvement.
>
> First, we would like to address your concern that this paper is itself not very novel. This criticism is based on the fact that we are proposing two algorithms that combine PSRO and Q-Mixing, and do not introduce either of these technologies within this paper. We can see how the experimental focus on using Q-mixing can give this impression. However (1) our approach does not require the use of Q-mixing, alternative methods for mixing pre-trained strategies could be used as the TransferOracle or OpponentOracle; (2) Our work is the first to use such a technology for combining pre-trained policies in PSRO, connecting two previously separate research directions; (3) we proposed two different methods, mixed oracles and mixed opponents. The question of how best to use Q-mixing or a similar technology for PSRO is important, and we make novel contributions to answering this question; (4) we empirically validate that these approaches achieve efficiency improvements in PSRO.
>
> We will call out these important contributions more clearly in the camera-ready version of our work.
>
> Thank you for bringing up a related work P2RO. Pipeline-PSRO (P2RO) is a variation of PSRO that has every epoch of PSRO train simultaneously against a moving target mixture of opponents. The focus of P2RO is not directly on reducing the simulation requirements, but instead the amount of elapsed time for the algorithm to complete by improving scalability. Therefore, P2RO will generally have a lower wall-clock time, but require the same or larger simulation budget then PSRO, and a substantially larger simulation budget than our proposed algorithms. Moreover, P2RO is altering PSRO at the “outer loop” (by attempting to run each epoch simultaneously), and as a result, the speed-up can also be applied to both Mixed-Opponents and Mixed-Oracles. Due to the focus of P2RO being on a different problem we do not think it makes sense as a baseline, but we will include this discussion in the related work.
>
> Thank you for the concrete feedback on the methodology section. We will expand on the methodology section to include more of a discussion on the OpponentOracle and how Q-Mixing is utilized.

---

> > ### Comment · AnonReviewer4 · 2020-11-17
> > **Thanks for the answer**
> >
> > Thank you for the clarification, especially for the comment on P2SRO, and the revision of the paper. After your feedback I have updated my score.

---

### Official Review · AnonReviewer3 · 2020-10-28
**Interesting work but exposition makes it hard to assess.**

**Rating:** 7
**Confidence:** 4

**Review:**

##########################################################################
Summary:

The paper provides an interesting approach to speeding up the convergence time of the Policy-Space Response Oracles framework by re-using the Q-functions of past best-responses to transfer knowledge across epochs.
##########################################################################
Reasons for score:

Overall, I am low confidence on my assessment of this paper due to the exposition in the algorithm section being relatively confusing. The experimental results are interesting which suggests that the method has value but there is key missing information on how the best response policies are constructed that make it difficult to assess the paper and lead to my not wanting to recommend its acceptance. I would highly recommend being more detailed in Sec. 3 to allow me to reassess the paper. I would certainly be willing to update my score if the paper was clearer to read.
 ##########################################################################Pros:

1. The experimental results on Leduc Poker are very speedy in terms of time-steps.

2. The idea of reusing the prior Q functions and just mixing them together rather than re-learning all of the policies is very good.


##########################################################################
Cons:

1. The algorithm boxes are so high level that I am struggling to understand how the algorithms work. I would not be able to implement it from reading the paper.

#########################################################################
Things that would improve readability:

- It would be nice in the algorithm boxes to connect Q-mixing to how the best policy is explicitly output. I was not able to understand how Q-mixing was connected to either Algorithm 2 or 3 and subsequently had difficulty following the paper.
- \lambda does not appear to be defined anywhere but appears in the Mixed Oracles algorithm box
- What is the OpponentOracle and the TransferOracle? They are defined in the algorithm boxes but are not clearly defined elsewhere.
- The specific example of RPS in section 3.2 does not provide useful intuition by going through the numerics, it may be more helpful to walk through a more high level description.
- It would probably be useful to move more of the experimental results to an appendix to leave room for the exposition of the algorithms.

---

> ### Author Response · Authors · 2020-11-13
> **Response to Reviewer #3**
>
> Thank you for taking the time to review our work and providing detailed feedback. We are glad that you found the underlying idea behind Mixed-Oracles to be good, and that you thought our experimental results were promising.
>
> Your feedback on our methods section is particularly helpful, we have edited Section 3 for clarity based on the suggestions you’ve offered in this review. In particular, we have a provided more detailed discussion of the algorithmic details in including the notation and functions called. To directly address your concern, the best-response for each method are computed as follows:
> - Mixed-Oracles: Recall in this algorithm we are interested in transferring best-response information across epochs of EGTA (limited to 2-player games). To accomplish this we exploit that PSRO only adds a single new policy to each player’s strategy-set during each epoch. As a result, during each epoch, we can utilize Deep RL to construct a best-response to the new pure-strategy generated from the previous epoch (we denote this with lambda). Now each player will have a set of best-responses to the other player’s individual policies (we denote the set with Lambda). However, the algorithm still must add a best-response to the current empirical game’s solution. This is where we introduce a general function “TransferOracle” which takes in the current solution and set of best-responses and produces a best-response to the provided opponent’s solution (the best-response to sigma_{-i}). In this study, we investigate Q-Mixing as our TransferOracle method. The Q-Mixing policy practically looks like a mixture-of-experts over several value-based policies.
>  - Mixed-Opponents: Recall in this algorithm we are modifying the best-response objective. In this case, we are attempting to aggregate a mixed-strategy into a single new policy. This will enable us to obtain two benefits (1) reduce the state-outcome variance induced by the mixed-strategy, and (2) leverage additional information about each opponent policy in the mixed strategy. To do this we introduce a general function “OpponentOracle” which has the same form as the “TransferOracle”. Similar to Mixed-Oracles, we utilize Q-Mixing to aggregate all of the opponent policies into a new opponent policy. This new policy serves as the best-response objective for each player when they are expanding their strategy-set.
>
> Addressing your readability concerns:
>  - Q-Mixing was used as both the TransferOracle and OpponentOracle. It took the respective set of policies and some distribution over the policies, and aggregated the information into a single policy (following the Q-Mixing algorithm).
>  - Thank you for pointing out this error. We have added this to the discussion in section 3. Lower-case lambda is a single best-response to the one of the policies in the opponent’s strategy-set, and upper-case lambda is the set of lambdas.
>  - Apologies again for the definition of OpponentOracle and TransferOracle being implicit. They are functions from strategy-sets and distributions onto single policies. Q-Mixing is an example of one such function, and is the method used in this work. This will be made explicit in section 3.
>  - We have edited section 3.2 to remove potentially distracting precision (we’ve kept the precise numerics as an appendix: we think that the example really arises in RPS is also important). The key idea here that “The problem is that there are already strategies in the empirical game that can defeat player 2's strategies, so the new strategy is not useful. This can be detected by inspecting player 2's Q-functions...”. Please let us know if the new phrasing is an improvement
>  - Thank you for the suggestion, we will shift things to the appendix if the space is necessary to correct the methodology.

---

> > ### Comment · AnonReviewer3 · 2020-11-16
> > **Much better! A few requests for more clarity.**
> >
> > Hi, most of my concerns have been alleviated by the clear rebuttal.
> >
> > However, I think it would still be helpful to readers to add a much more detailed version of the algorithm blocks to the appendix. Historically (from having been in reading groups where PRSO and related variants have been discussed), I have observed that readers have trouble understanding the algorithms. There's little downside to doing this and the potential of the paper being significantly more readable.
> >
> > Finally, I think the training objective under Q-mixing is still somewhat unclear and it would be useful to add the explicit loss / objective that is being used here since there is still remaining space.
> >
> > I have updated my score and am open to updating it further if increasing clarity is added.

---

> > > ### Author Response · Authors · 2020-11-17
> > > **Response to Additional Request for Clarity**
> > >
> > > Thank you for getting back promptly. We are glad to hear that our posted changes have resolved most of your concerns about the clarity in our work.
> > >
> > > We have added an extensive section to the supplementary material that rigorously explains the details encountered in PSRO and our two proposed variants. If you have any additional suggestions on how we may improve this exposition please do not hesitate to let us know!
> > >
> > > Moreover, we have added to this new supplementary material section a sub-section devoted to Q-Mixing. This sub-section practically walks through how Q-Mixing is used as a TransferOracle and/or OpponentOracle. Notably, the version of Q-Mixing we are using does not use any loss or objective functions (instead it functions more as an ensemble of policies). The BR calculations in PSRO, Mixed-Opponents, and Mixed-Oracles all follow the same loss functions with different objectives (determined by the opponent or distribution of opponents that are being faced).
> > >
> > > Please let us know if there's anything else that you found unclear, and we will happily work with you to resolve the issues.

---

> > > > ### Comment · AnonReviewer3 · 2020-11-23
> > > > **Minor comments**
> > > >
> > > > Hi,
> > > >
> > > > Thank you for doing this! I think this makes the paper a lot clearer to read.
> > > > A few minor comments:
> > > > - In algorithm 4, it seems like there's some superscript error.
> > > > - In the mixed oracles section there is a small grammar error ("that we are responding too instead of "that we are responding to")
> > > >
> > > > I'll be upgrading my score. However, I think given the importance of the appendix for providing clarity to this paper, it may be worthwhile to direct the reader there explicitly. Additionally, if you wind up with extra page space, it may be worth using it to directly explain how Q-mixing is used. Currently this section is in the appendix but the paper became a lot clearer after reading it.

---

### Decision · Program_Chairs · 2021-01-07
**Final Decision**

**Decision:**

Accept (Spotlight)

**Comment:**

This paper proposes a method to improve the convergence time of PSRO. The paper was well received by all reviewers and is likely to be of interest to a similar sub-community within ICLR, but may be of less relevance to the wider community not focused on multi-agent learning.

A number of issues were raised by reviewers regarding the clarity of the originally submitted version of the paper. I encourage the authors to consider all constructive feedback given and revise the paper to maximise its impact. This will be of particular help in reaching a wider audience than those with pre-existing experience with the methods this work builds on.